# Priority-Aware Pathological Hierarchy Training for Multiple Instance Learning

**Abstract.** Multiple Instance Learning (MIL) is increasingly being used as a support tool within clinical settings for pathological diagnosis decisions, achieving high performance and removing the annotation burden. However, existing approaches for clinical MIL tasks have not adequately addressed the priority issues that exist in relation to pathological symptoms and diagnostic classes, causing MIL models to ignore priority among classes. To overcome this clinical limitation of MIL, we propose a new method that addresses priority issues using two hierarchies: vertical *inter-hierarchy* and horizontal *intra-hierarchy*. The proposed method aligns MIL predictions across each hierarchical level and employs an implicit feature re-usability during training to facilitate clinically more serious classes within the same level. Experiments with real-world patient data show that the proposed method effectively reduces misdiagnosis and prioritizes more important symptoms in multiclass scenarios. Further analysis verifies the efficacy of the proposed components and qualitatively confirms the MIL predictions against challenging cases with multiple symptoms.

**Keywords:** Multiclass Priority · Class Hierarchy · Multiple Instance Learning.

## 1 Introduction

The exponential increase in the demand for pathological diagnoses after the COVID-19 pandemic has significantly burdened a limited number of pathological specialists [4, 2]. At the same time, the deep learning (DL) community has actively pursued alleviating this workload by developing automated diagnostic models to assist pathological decision making [7]. Recently, Multiple Instance Learning (MIL), which uses only weak labels at the Whole Slide Image (WSI) level for model training, not pixel-level annotations by experts, has emerged as the golden standard in digital pathology diagnosis [22].

Although MIL offers promising results for expert assistance in clinical settings, it reveals shortcomings in multiclass scenarios, as most MIL studies have been conducted in binary settings [18, 21]. Unlike binary classification, a multiclass task commonly involves a hierarchy because lower-level classes can be organized into groups of higher levels [1], potentially reflecting priorities or different clinical urgencies between those higher groups. The DL community has

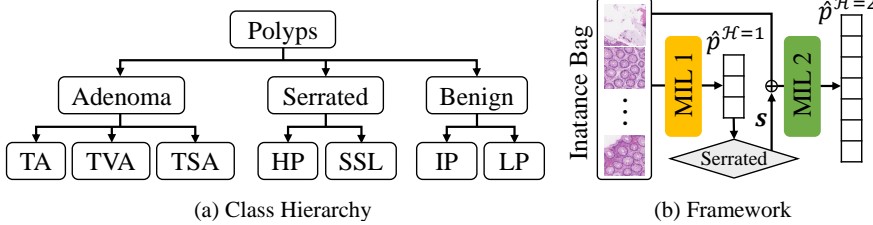

Fig. 1: (a) A diagram illustrating class relationships and hierarchy. We denote the structure from root to leaves as $\mathcal{H} = 0$ to $\mathcal{H} = 2$. (b) The proposed framework offers a two-phase design, which is trained end-to-end manner.

made several attempts to leverage hierarchy, such as loss-centric methodologies, which penalize predictions based on class relationships [1, 5]. Structure-based methods try to establish these class relationships within the framework [17], graphs [3], and hyperbolic space [15]. The underlying objective of these inter-hierarchy approaches is to prevent networks from making critical fine-level errors in classification, which correspond to type II errors in medical field (*e.g.*, A model might mistake a stage of tumor, but should not confuse a cancerous cell with a normal one).

Despite previous attempts to address the hierarchy issues, the inherent properties of WSIs impose limitations on conventional multiclass hierarchy approaches. Although WSI training uses only one label, clinical inference often involves multiple symptoms, which requires pathologists to identify the most urgent problem [20, 19]. As models are trained with the assumption of a strict label, they are prone to concentrate on the most probable class, rather than the most hazardous sign [11, 10]. We refer to this issue, ignoring priority within the horizontal hierarchy, as an intra-hierarchy problem.

We address the hierarchy issues in multiple ways. For inter-hierarchy, we utilize a probability alignment term between each hierarchy. Concurrently, we propose a probability adjustment that allows the coarse-grained hierarchy to influence the predictions of the fine-grained hierarchy. We also present an implicit feature remix to handle the intra-hierarchy problem. Given that the input of MIL is a set of multiple instances, we implicitly train class priority by mixing instances from two samples. We have confirmed that it enables the model to focus on the more urgent class in a complex test set where two cases are mixed. The proposed framework flexibly employs MIL architectures and leverages multimodal data. Experiments conducted on real-world clinical data show that the proposed method outperforms the extant methods while properly respecting multi-class hierarchies. Through ablation studies, we confirm the contribution of each component. Additional qualitative evaluations examine the predictions of the proposed methodology on challenging diagnostic images.

**Related Work** There are several class hierarchy-aware classifiers. DeViSE [8] optimizes cosine similarity between image embeddings from pretrained visual

Table 1: Data distribution over the classes. The values in parentheses represent the number of extra test samples.

|            | TA       | TVA    | TSA    | HP    | SSL     | IP | LP | $\sum$    |
|------------|----------|--------|--------|-------|---------|----|----|-----------|
| Train      | 317      | 232    | 300    | 257   | 130     | 99 | 266| 1,601     |
| Validation | 69       | 51     | 65     | 55    | 29      | 21 | 57 | 347       |
| Test       | 164(95)  | 57(6)  | 84(18) | 64(8) | 84(55)  | 21 | 57 | 531(182)  |

models and label embeddings from Word2Vec [14]. Bertinetto et al. [1] introduced hierarchy-sensitive loss adaptations to reduce hierarchical distance in top-$k$ predictions while trading off top-1 accuracy. Chang et al. [5] addressed how coarse class cross-entropy loss degrades fine-grained accuracy by partitioning the feature space to disentangle coarse and fine-grained features. Garg et al. [9] proposed a feature learning method that considers class hierarchies, using Jensen-Shannon divergence and geometric constraints to train hierarchical semantic organization. While previous studies exploited class hierarchies in a coarse-to-fine manner, the lack of explicit class priority specification within hierarchies makes hierarchical approaches worth exploring, particularly for multiclass clinical WSI settings.

## 2    Method

### 2.1    Data Description

We use 2,297 digital WSIs originated from patients in a real-world clinical setting of `AnonymousCenter`[1], which comprises a total of seven classes: tubular adenoma (TA), tubulovillous adenoma (TVA), traditional serrated adenoma (TSA), hyperplastic polyp (HP), sessile serrated lesion (SSL), inflammatory polyp (IP), and lymphoid polyp (LP). These classes are organized into three coarser categories, as illustrated in Fig. 1 (a). Among them, Adenoma is paramount due to its potential for malignant transformation. Serrated is of secondary importance, necessitating more detailed diagnosis into SSL and HP. Each WSI has a Subsite indicating specimen location: `Proximal` for near the oral cavity, `Distal` for near the anus, `UNKNOWN` otherwise. We convert it into a three-dimensional one-hot vector $\mathbf{s}$. This clinical dataset, comprising WSIs each with a single symptom, was split into training, validation, and test sets at a 0.7:0.15:0.15 ratio. In addition, we have incorporated an additional 182 complex samples (see Table. 1), which contain two or more symptoms, into the test set, to assess the proposed method's performance in challenging real-world multi-symptom conditions.

### 2.2    Proposed Two-phase Framework

Fig. 1 (b) shows the two-phase framework we propose in consideration of the *intra-* and *inter-* hierarchical relationships. A WSI $X_i$ is separated into $n(X_i)$

---

[1] This study was performed in line with the principles of the Declaration of Helsinki. Approval was granted by the Ethics Review Board (∗-IRB-∗-∗) and (∗-IRB-∗-∗∗).

patches $\{x_{i,1}, \cdots, x_{i,n(X_i)}\}$, and a pre-trained feature extractor outputs the corresponding instance bag $\mathcal{B}_i = \{z_{i,1}, \cdots, z_{i,n(X_i)}\}$. The $\mathcal{B}_i$ is fed into each $\mathcal{H}$ MIL, $f_{\theta_1}(\cdot)$ and $f_{\theta_2}(\cdot)$. We denote the softmax outputs of $f_{\theta_1}(\cdot)$ and $f_{\theta_2}(\cdot)$ as $\hat{p}^{\mathcal{H}=1} \in \mathbb{R}^3$ and $\hat{p}^{\mathcal{H}=2} \in \mathbb{R}^7$, respectively. Observing that pathologists closely examine the acquisition site in the diagnosis of HP and SSL, we concatenate $\mathbf{s}$ with the input to feed into $f_{\theta_2}(\cdot)$ if $\mathrm{argmax}_c(\hat{p}^{\mathcal{H}=1})$ is Serrated. Consequently, each hierarchy MIL is trained in an end-to-end manner with the proposed framework using the following cross-entropy term:

$$\mathcal{L}_{CE} = -\frac{1}{2} \sum_{h \in \mathcal{H}} \sum_{c \in \mathcal{C}^{\mathcal{H}=h}} y_c^{\mathcal{H}=h} \log(\hat{p}_c^{\mathcal{H}=h}) \tag{1}$$

, where $\mathcal{C}^{\mathcal{H}}$ indicates the classes that are allocated in $\mathcal{H}$.

### 2.3   Inter-Hierarchy Alignment

Although hierarchical MILs predict a different number of classes, they share the same input. That is, given that both MILs evaluate the same samples, the lower-level probability distribution, aggregated to match the higher-level classes, should ideally match the higher-level probability distribution. Inspired by this motivation and [9], we enforce $\hat{p}^{\mathcal{H}=1}$ and $\hat{p}^{\mathcal{H}=2}$ to be aligned:

$$\mathcal{L}_{IHA} = \mathrm{JS}(\hat{p}^{\mathcal{H}=1} || \dot{p}^{\mathcal{H}=1}) = \frac{1}{2}\left(\mathrm{KL}(\hat{p}^{\mathcal{H}=1} || m) + \mathrm{KL}(\dot{p}^{\mathcal{H}=1} || m)\right) \tag{2}$$

, where $m = \frac{1}{2} \times (\hat{p}^{\mathcal{H}=1} + \dot{p}^{\mathcal{H}=1})$. JS and KL denote Jensen-Shannon Divergence and Kullback-Leibler Divergence, respectively. We perform the following operation to obtain the average distribution $m \in \mathbb{R}^3$ and the aligned probability $\dot{p}^{\mathcal{H}=1} \in \mathbb{R}^3$ from $\mathcal{H} = 2$ to $\mathcal{H} = 1$:

$$\dot{p}_c^{\mathcal{H}=1} = \sum_{c' \subset c} \hat{p}_{c'}^{\mathcal{H}=2}, \text{ where } c \in \mathcal{C}^{\mathcal{H}=1} \text{ and } c' \in \mathcal{C}^{\mathcal{H}=2} \tag{3}$$

### 2.4   Upper-Hierarchy-Dependent Probability

Classifying the three classes of $\mathcal{H} = 1$ is a simpler task than the seven classes of $\mathcal{H} = 2$. In other words, if $f_{\theta_1}(\cdot)$ and $f_{\theta_2}(\cdot)$ refer to each other, it is reasonable to do so from the coarse to the fine level. Therefore, we adjust the probabilities $\hat{p}^{\mathcal{H}=2}$ of $f_{\theta_2}(\cdot)$, so that it aligns with the predictions of the $f_{\theta_1}(\cdot)$ while also allowing for some dependence:

$$\tilde{p}_c^{\mathcal{H}=2} = \begin{cases} \hat{p}_c^{\mathcal{H}=2} \times \hat{p}_{c'}^{\mathcal{H}=1} & \text{, if } c \subset c' \\ \hat{p}_c^{\mathcal{H}=2} & \text{, otherwise.} \end{cases} \tag{4}$$

$$\mathcal{L}_{UHD} = \mathrm{KL}(||\tilde{p}^{\mathcal{H}=2}||_1 || y^{\mathcal{H}=2}) \tag{5}$$

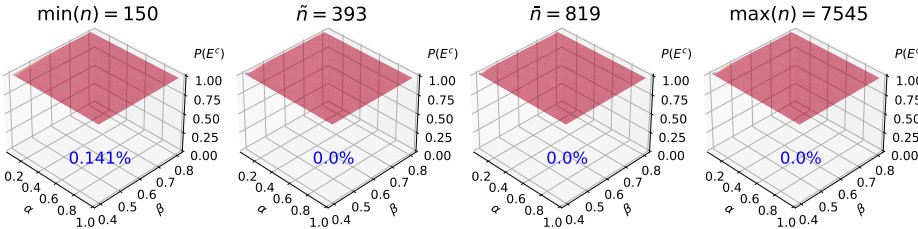

Fig. 2: $P(E^c)$ is visualized for the median, average, and maximum counts of $n = |\mathcal{B}_i|$. In each plot, the blue percentage indicates the proportion of cases with a probability of 99% or less out of feasible events.

### 2.5 Implicit Feature Remix for Intra-Hierarchy

We still have the second component of the class hierarchy: Intra-hierarchy. Given that a $\mathcal{B}$ is a collection of multiple instances, a random proportion $\beta \sim$ Uniform$(0.4, 0.8)$ of instances is sampled from $\mathcal{B}_i$ and mixed into the $1 - \beta$ proportion of $\mathcal{B}_j$ to synthesize bag $\mathcal{B}_{i+j}$, where $\mathcal{B}_i$ has higher priority than $\mathcal{B}_j$ within the same $\mathcal{H}$ (*e.g.*, TA and LP). We perform feature remixing only when $\mathcal{B}_i$ has at least 150 instances to create a distinguishable synthesized sample. Here, a valid concern is the possibility that the following event $E$ occurs: $E \coloneqq \{$*no crucial instance for diagnosis from $\mathcal{B}_i$ are mixed into $\mathcal{B}_{i+j}$*$\}$. However, contrary to our concerns, if we assume $\mathcal{B}_i$ contains a proportion of $\alpha \geq 0.05$ instances exhibiting symptoms, then event $E$ is rarely to occur (*i.e.*, complementary set $E^c$):

$$P(E^c) = \begin{cases} 1 - \frac{(n-n\alpha)C_{n\beta}}{nC_{n\beta}} & \text{, if } \alpha + \beta < 1, \\ 1 & \text{, if } \alpha + \beta \geq 1. \end{cases} \tag{6}$$

For clearer comprehension, we plot the form of Equ. 6 in Fig. 2. This shows that the synthesized $\mathcal{B}_{i+j}$ contains instances that exhibit symptoms of a higher intra-hierarchy class.

The label of $\mathcal{B}_{i+j}$ can be $y_i^{\mathcal{H}}$ of the higher intra-hierarchy bag $\mathcal{B}_i$, however, to utilize the benefits of label softening [6], $k$-th dimension of the smoothed label vector $y_{i+j}^{\mathcal{H}}$ is defined as:

$$y_{i+j,k}^{\mathcal{H}} = \begin{cases} \tilde{r}_k/(\tilde{r}_i + \tilde{r}_j), \text{ if } k \in \{i, j\} \\ 0 \qquad \text{, otherwise.} \end{cases}$$

$$, \text{ where } \begin{cases} \tilde{r}_i = r^{1/\tau} \\ \tilde{r}_j = (1-r)^\tau \end{cases} \text{ and } r = \frac{\beta \times |\mathcal{B}_i|}{\beta \times |\mathcal{B}_i| + (1-\beta) \times |\mathcal{B}_j|} \tag{7}$$

, where $\tau$ is the smoothing factor. The condition for $\bar{r}_i$ and $\bar{r}_j$ is designed to make the class of $\mathcal{B}_i$ dominant in $y_{i+j}^{\mathcal{H}}$.

The proposed hierarchical MIL framework is trained by the term $\mathcal{L} = \lambda_1 \mathcal{L}_{CE} + \lambda_2 \mathcal{L}_{IHA} + \lambda_3 \mathcal{L}_{UHD}$, where each $\lambda_{1:3}$ are hyper-parameters.

Table 2: The performance comparisons of alternative hierarchy-aware methods. The values in parentheses indicate standard deviation.

| | TransMIL [18] | | | | | |
|---|---|---|---|---|---|---|
| Method | $\mathcal{H}=1$ | | | $\mathcal{H}=2$ | | |
| | Accuracy | AUROC | Recall | Accuracy | AUROC | Recall |
| CE | - | - | - | 0.866(0.008) | 0.987(0.001) | 0.933(0.020) |
| Weighted CE (5:3:2) | - | - | - | 0.870(0.012) | 0.987(0.001) | 0.933(0.020) |
| Weighted CE (7:2:1) | - | - | - | 0.851(0.011) | 0.986(0.002) | 0.916(0.027) |
| HXE ($\alpha=0.1$) [1] | 0.916(0.009) | 0.985(0.002) | 0.908(0.013) | 0.876(0.017) | 0.987(0.002) | 0.948(0.008) |
| HXE ($\alpha=0.3$) [1] | 0.912(0.010) | 0.986(0.002) | **0.937(0.027)** | 0.875(0.007) | 0.988(0.001) | 0.941(0.017) |
| Soft Labels ($\beta=5$) [1] | 0.908(0.009) | 0.982(0.004) | 0.914(0.012) | 0.882(0.012) | 0.985(0.002) | 0.953(0.012) |
| Soft Labels ($\beta=10$) [1] | 0.918(0.014) | 0.981(0.011) | 0.929(0.022) | 0.868(0.009) | 0.982(0.002) | 0.933(0.017) |
| Chang et al. [5] | 0.920(0.009) | 0.981(0.003) | 0.924(0.009) | 0.872(0.007) | 0.985(0.003) | 0.941(0.013) |
| HAF [9] | 0.865(0.045) | 0.960(0.003) | 0.910(0.042) | 0.869(0.015) | 0.986(0.002) | 0.940(0.022) |
| Ours | **0.922(0.009)** | **0.989(0.001)** | 0.927(0.029) | **0.898(0.006)** | **0.990(0.002)** | **0.972(0.008)** |
| | DTFD-MIL [21] | | | | | |
| CE | - | - | - | 0.860(0.014) | 0.986(0.002) | 0.918(0.016) |
| Weighted CE (5:3:2) | - | - | - | 0.871(0.012) | 0.987(0.001) | 0.933(0.020) |
| Weighted CE (7:2:1) | - | - | - | 0.850(0.019) | 0.984(0.002) | 0.896(0.014) |
| HXE ($\alpha=0.1$) [1] | 0.922(0.023) | 0.985(0.002) | 0.911(0.053) | 0.875(0.003) | 0.987(0.001) | 0.934(0.012) |
| HXE ($\alpha=0.3$) [1] | 0.926(0.009) | 0.987(0.001) | 0.924(0.020) | 0.863(0.003) | 0.987(0.001) | 0.924(0.006) |
| Soft Labels ($\beta=5$) [1] | 0.923(0.011) | 0.986(0.003) | 0.925(0.015) | 0.874(0.007) | 0.980(0.002) | 0.927(0.010) |
| Soft Labels ($\beta=10$) [1] | 0.915(0.010) | 0.983(0.004) | 0.916(0.022) | 0.866(0.008) | 0.984(0.002) | 0.930(0.018) |
| Chang et al. [5] | 0.941(0.008) | 0.987(0.002) | 0.944(0.027)) | 0.879(0.016) | 0.987(0.004) | 0.947(0.016) |
| HAF [9] | 0.894(0.023) | 0.976(0.006) | 0.865(0.042) | 0.862(0.012) | 0.986(0.003) | 0.916(0.020) |
| Ours | **0.948(0.007)** | **0.991(0.001)** | **0.955(0.013)** | **0.892(0.014)** | **0.991(0.001)** | **0.970(0.010)** |

## 3   Experiment

### 3.1   Implementation Details

**MIL Architectures** We utilize two state-of-the-art MIL architectures: TransMIL [18] and DTFD-MIL [21]. TransMIL optimizes computation while capturing more advanced inter-instance relationships. DTFD-MIL conducts double-tier distillation by resampling the input into pseudo-bags. For DTDF-MIL, we adopt Aggregated Feature Selection, which typically yields superior performance.

**Training Settings** We set the $\tau$ as 15 and $\lambda_1 = \lambda_2 = \lambda_3 = 1$. We selected $\times 256$ size patches from the 1MPP of WSIs using the Otsu algorithm [16], then transformed them into individual instances with a pre-trained feature extractor [12]. We trained the model using Adam optimizer [13] with betas of $(0.9, 0.999)$ and a learning rate of $1e-4$. All experiments were carried out with fixed seeds on a single NVIDIA® A6000 with 48GB of memory.

**Comparison Methods** We set cross-entropy (CE) and weighted CE, which explicitly trains for the importance of classes, as the baseline. Hierarchical CE (HXE) and soft labels [1], Chang et al. [5], and hierarchy-aware feature (HAF) [9] were selected as comparison methods that can handle coarse-to-fine hierarchy. For fair comparisons, we repeated all experiments with the optimized hyperparameters for each method, reporting the mean and standard deviation.

**Evaluation Metrics** We evaluate the performance at each $\mathcal{H}$ with Accuracy, AUROC, and Recall scores. In particular, the recall measure used here is based on a binary metric, where the positive class is defined as Adenoma or any of its subclasses.

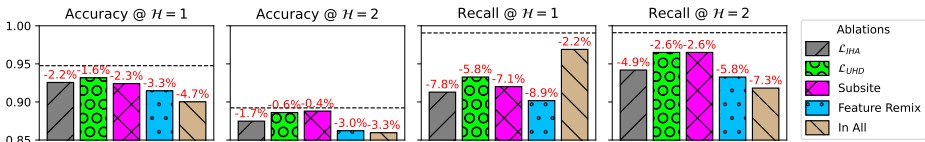

Fig. 3: Ablation results on the core components of the proposed method using DTFD-MIL. Dashed lines indicate the fully equipped model's performance.

### 3.2  Quantitative Results

Table. 2 presents the results of running various hierarchy-aware methods against the test data. Applying weighted CE with 5:3:2 weights improves the baseline in both MIL structures. However, it also shows that excessively high weights for certain classes can reverse this gain, making performance worse than the baseline, highlighting that methods requiring explicit parameterization necessitate domain expertise and considerable empirical search.

Moreover, HXE [1] had difficulty leveraging its advantages in the minimal depth hierarchy because its conditional term operates with limited information, which hinders the differentiation of importance of the class. Consistent performance gains are observed across all comparison groups with the weak soft-labels [1] (*i.e.*, $\beta = 5$). The HAF [9] results reveal that hierarchical feature alignment is not critical for MIL. This phenomenon can be attributed to the representational disparity: linear networks exhibit limited interaction while attention-based MIL captures nuanced feature correlations, which are not adaptable across the hierarchies. Upon the results of Chang et al. [5], it shows remarkable performance at $\mathcal{H} = 1$ compared to other methods, due to training that emphasized coarser information through initial epochs. Finally, our proposed approach yielded superior performance compared to other methods, without exception. Not only did it ensure high accuracy, but also showed the lowest type II error rates, which is critical in medical domain. The findings indicate that our approach provides a suitable solution for real-world clinical WSIs, given their vertical inter-class hierarchy and diagnostic priority at the same level.

### 3.3  Further Analysis

**Ablation Study** We have conducted an ablation study to understand each component's effect on performance. Results of removing $\mathcal{L}_{IHA}$, $\mathcal{L}_{UHD}$, subsite **s**, feature remix, and all components are shown in Fig. 3. Each removal caused performance degradation, with excluding all components showing the worst results. Removing subsite at $\mathcal{H} = 2$ also affected precision at $\mathcal{H} = 1$, indicating probability alignment impacts $\mathcal{H} = 1$ performance. Feature remix ablation resulted in the most substantial degradation, highlighting its importance for WSIs with multiple symptoms. Moreover, regarding the feature remix component, the concurrent increase in both recall and accuracy suggests that the improved recall is derived from precise diagnoses, not simply over-predicting positive cases.

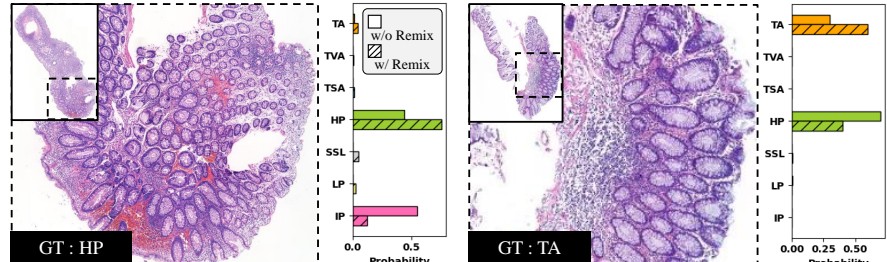

Fig. 4: Quantitative investigation into cases with mixed symptoms. We plot the $\hat{p}^{\mathcal{H}=2}$ of models trained with feature remixed samples against those of models trained without, shown with the corresponding WSIs.

**Analysis on Intra-Hierarchy** To understand how the model performs against challenging cases, we have examined whether the model prioritizes the most urgent class when two or more cases are mixed within a WSI. The left tissue in Fig. 4 presents an HP with substantial IP mixture. Without intra-hierarchy training, the MIL model predicts IP with greater confidence than HP, simply due to the symptom area. In contrast, a model that implicitly learns the diagnostic precedence of HP over IP predicts the case with the more serious diagnosis. The tissue on the right is a sample that pathologists diagnose as TA, but previous MIL approaches classified it as HP. Although a small area of TA is observed in the magnified view, it is expected to have a higher probability because it is more urgent than HP. Implicit feature remix prioritizes the class with higher precedence when multiple classes are present in an instance bag.

## 4    Conclusion

Our research aims to solve the constraints that currently impede the successful implementation of multiclass WSI MIL in real-world clinical settings. With the formulation of the class hierarchy in two alternative ways, our proposed method offers key components effective for each. Inter-hierarchy alignment of predictions across the vertical hierarchy contributes to improved performance. The predictions of the fine-grained hierarchy are influenced by the coarse-grained hierarchy, thus having fine probabilities adjusted to ensure consistency with the coarse. Implicit feature remix allows the model to understand diagnostic urgency in mixed-symptom inference environments, relying solely on a weak label. The results of the experiment have shown that the feature remix improved the quantitative performance and allowed it to focus on more prioritized diagnoses. Furthermore, we have explored the applicability of the class hierarchy, a novel concept in MIL, by comparing it with various methods. Our proposed method mitigates the challenges of multiclass MIL diagnosis of previous approaches, broadening its applicability to practical use in clinical settings.

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
