# OpenReview forum: "Priority-Aware Clinical Pathology Hierarchy Training for Multiple Instance Learning"
_MICCAI.org/2025/Workshop/MSB_EMERGE — MSB EMERGE 2025 Oral_

### Official Review · Reviewer_gfGa · 2025-07-04

**Recommendation:** 5
**Confidence:** 4

**Clarity:**

The paper is clear and well-written, with minor areas for improvement in clarity

**Feedback:**

- Fig. 4, how did you pick the samples? Do you have an idea on how to quantify this aspect to confirm this for the whole dataset?
- Fig 2 is unclear: Further description of the figure in the title is needed
- Lambdas are introduced in the loss and then all set to 1 after.

**Justification:**

The authors proposed a methods to tackle an important limitations of MLI and demonstrate superior performance against an extensive comparison.

**Reproducibility:**

Sufficient amount of details available for reproducing the main results, but open access is not provided to source code and/or data

**Strengths:**

-	Extensive comparison of the methods with two state-of-the-art MIL and 8 different methods, and 3 evaluation metrics.
-	Relevant ablation study conducted showing the benefit of their methods
-	The experiment setup is clear and well described.

**Summary:**

In this paper, the authors proposed a two-step framework to address the priority issues of Multiple Instance Learning methods, i.e. models ignore priority among classes, when used for pathological diagnosis. The framework considers horizontal intra-hierarchy and vertical inter-hierarchy. They demonstrate the benefits of their methods in an extensive benchmark and also the benefits of each component of the proposed method in an ablation study.

**Weaknesses:**

-	Reproducibility: Experiments were done only on a private dataset

---

### Official Review · Reviewer_kwsP · 2025-07-05

**Recommendation:** 4
**Confidence:** 3

**Clarity:**

The paper is generally clear but has some clarity issues that could be addressed with moderate revision

**Feedback:**

- It would be great to clarify hierarchy terms with examples or diagrams early in the paper.
- Provide clearer implementation details and ablation studies especially for the hyperparameters.
- Figure 1: It's somehow abstract, lacking details on how patches are processed or how loss terms interact. Add annotations to the diagram to clarify data flow and loss computation.

**Justification:**

The paper offers a good contribution to MIL in pathology by addressing class priority, with robust evaluation and clinical relevance.

**Reproducibility:**

Some amount of details available for reproducing the main results, and open access details are unclear

**Strengths:**

- I liked that the paper addresses a critical gap in MIL by prioritizing urgent classes in multiclass settings.
- The authors use a real-world dataset with 2,297 WSIs, including 182 complex test cases, and compares against multiple baselines.
- The proposed method reduces type II errors, crucial for avoiding missed serious diagnoses.

**Summary:**

The paper proposes a novel Multiple Instance Learning (MIL) framework for multiclass pathological diagnosis, addressing class priority through a two-phase approach handling vertical (inter-hierarchy) and horizontal (intra-hierarchy) relationships. It aligns predictions across hierarchies and uses implicit feature remixing to prioritize clinically urgent classes.

**Weaknesses:**

- Vertical inter-hierarchy and horizontal intra-hierarchy lack intuitive explanations. I personally didn't know about these and had to read a bit.
- Dataset from a single center; no discussion of applicability to other datasets or pathologies. So how would the method generalize?
- No justification for hyperparameter choices especially lambdas which are all set to 1.
- Multiple loss terms and feature remixing may hinder practical adoption.

---

### Official Review · Reviewer_26z7 · 2025-07-08

**Clarity:** The paper is exceptionally well-writt…
**Recommendation:** 5
**Confidence:** 4

**Feedback:**

- In several instances (e.g., Equation 1), commas are not placed immediately after displayed equations, but instead appear on the line below. It is standard practice to include punctuation on the same line as the equation.
- It would improve clarity if the symbol “:=” were used to indicate the definition of concepts, probabilities, vectors, and other expressions. This notation clearly signals to the reader that a term is being introduced or formally defined, rather than merely stated, e.g. Equations 3 and 7.
- Equation 5 appears to be isolated. This may be an oversight concerning punctuation marks.
- The variable representing the hierarchy $\mathcal{H}$ is currently defined only in Figure 2. For completeness and clarity, it would be helpful to introduce and define this variable explicitly in the main text as well.

**Justification:**

This paper addresses a relevant problem: enhancing MIL performance in multiclass scenarios leveraging class priority (inter- and intra-hierarchies). I found the paper to be methodologically sound and clearly presented. The method has improved performance over baseline methods.

**Reproducibility:**

Sufficient amount of details available for reproducing the main results, but open access is not provided to source code and/or data

**Strengths:**

- The paper tackles a relevant problem: enhancing MIL performance in multiclass scenarios.

- The manuscript is neatly written and easy to follow:
1. The introduction and related work sections are well-motivated, providing a clear context and rationale for the proposed approach.
2. The methodology is precisely formulated, with each step logically presented and easy to understand.
3. The experimental section is comprehensive and well-organized, making it easy to interpret the results.
4. The used figures complement the explanations well.

- The paper provides a precise description of the dataset, including the "construction" of intra- and inter-hierarchies. The explanation of class priority is well-aligned with clinical reasoning.

- The method is rigorously evaluated using multiple metrics (Accuracy, AUROC, Recall), and the inclusion of an ablation study underscores the contribution of each component of the method

**Summary:**

The authors introduce a novel Multiple Instance Learning (MIL) framework for multiclass pathological diagnosis using Whole Slide Images. Their method addresses a key limitation of existing hierarchical approaches—typically constrained to coarse-to-fine strategies—by explicitly leveraging and modeling class priority. To overcome this, they propose a dual-hierarchy strategy comprising vertical inter-hierarchy and horizontal intra-hierarchy structures. The proposed framework is evaluated using Accuracy, AUROC, and Recall, with results demonstrating improved performance over baseline methods, as presented in Table 2.

**Weaknesses:**

I have no major concerns or weaknesses to report.